# Polystyrene-Impregnated Glulam Resistance to Subterranean Termite Attacks in a Laboratory Test

**DOI:** 10.3390/polym14194003

**Published:** 2022-09-24

**Authors:** Yusuf Sudo Hadi, Dede Hermawan, Imam Busyra Abdillah, Mahdi Mubarok, Wa Ode Muliastuty Arsyad, Rohmah Pari

**Affiliations:** 1Department of Forest Products, Faculty of Forestry and Environment, IPB University (Bogor Agricultural University), Kampus IPB Darmaga, Bogor 16680, Indonesia; 2Center for Standardization of Sustainable Forest Management Instruments, Ministry of Environment and Forestry, Bogor 16610, Indonesia; 3Research Center for Biomass and Bioproducts, National Research and Innovation Agency (BRIN), Bogor 16610, Indonesia

**Keywords:** fast-growing tree, polystyrene glulam, resistance class, subterranean termites

## Abstract

This study aimed to enhance tropical fast-growing tree species’ resistance to subterranean termite (Coptotermes curvignathus) attacks through the manufacturing of polystyrene glued-laminated timber (glulam). Three young tropical wood species, namely manii (Maesopsis eminii), mangium (Acacia mangium), and rubber-wood (Hevea brasiliensis), were cut into laminae. After drying, the laminae were impregnated with styrene monomer, then polymerized using potassium peroxydisulfate as a catalyst and heat. The polystyrene-impregnated laminae were constructed using isocyanate glue and a cold press for three-layered glulam. Untreated or control glulam and solid wood specimens were also prepared. The specimens of each wood species and wood products (solid wood, control glulam, and polystyrene glulam) were exposed to the termite in a laboratory test according to Indonesian standards. The results showed that mangium wood had better resistance to the termite attack than manii and rubber-wood, with both of those woods performing the same. Among the wood products, the glulams were equal and had higher resistance to the termite attack than solid wood. To enhance the termite resistance of polystyrene glulam, we suggest that the polymer loading of polystyrene on each lamina should be increased. In our evaluation of the products’ order of priority, polystyrene glulam emerged as performing best towards subterranean termites attack.

## 1. Introduction

The demand for wooden products is continuously increasing every year. In the past decade, Indonesian log production increased by 2% each year, and currently, 87% of log production is from plantation forests [1,2]. In 2020, log production reached 61 million m^3^, and mangium (Acacia mangium) wood from plantation forests was dominantly produced, representing 52.6% [3]. Wood from plantation forests usually has a small diameter and low wood resistance, and it is especially susceptible to subterranean termite attacks. Previous studies estimated that the economic losses caused by termite attacks have reached 1 billion USD [4,5]. To overcome this issue, numerous methods have been developed for improving the wood resistance and utilizing small-diameter logs. Those methods include heat treatment, impregnation, and chemical modification [6,7,8,9,10,11,12].

Over the last decade, research on non-toxic biocide has been intensively conducted to find environmentally friendly products. Rowell [13] stated that acetylation, a well-known, non-toxic wood-modification approach, of yellow pine could improve the wood resistance to Gloeophyllum trabeum brown rot decay. The durability of this acetylated wood in ground stake testing after 10 years was also reported [14]. Meanwhile, Huaxu et al. [15] noted that citric acid-bonded rubber-wood particle board has significantly better fungal and termite resistance than urea-formaldehyde (UF)-bonded particle board. Furthermore, some studies used hydrothermal treatment to improve wood’s resistance to biodeteriorating agents [16].

In studying the utilization of wood from the plantations mentioned above, some researchers have reported that the small-diameter logs that were made into glued-laminated timber (glulam) were not different from solid wood in terms of physical and mechanical characteristics. Yet, compared to solid wood, the glulam had higher values for the modulus of rupture (MOR), modulus of elasticity (MOE), and hardness, but lower shear strength [17]. Moreover, impregnated polystyrene wood had increased resistance against subterranean termites [18]. 

Hence, these studies revealed that wood products with high termite resistance must be developed with a combined modification method. Among several modification methods, impregnation using polystyrene can be prescribed. This polystyrene may be recovered from waste plastic products [19,20]. Accordingly, it seems this treatment has the possibility to reduce the end-product costs [21].

Wood modification using the impregnation of polystyrene increased wood’s resistance to subterranean termites by 80% when exposed in a field test [22]. Furthermore, after a one-year exposure in the field, polystyrene wood had a weight loss of only 20%, while the untreated wood totally failed [23], and in another research, untreated sugi (Cryptomeria japonica) wood had a weight loss of 43.8% after exposure to subterranean termites in a laboratory test, while polystyrene-impregnated wood had only a 6.8% loss [18].

Polystyrene glulam can be constructed from some plies of polystyrene-impregnated laminae using adhesive and pressure. Hadi et al. [18] reported on some physical and mechanical properties of polystyrene glulam compared to untreated glulam, proposing that they were no different in terms of color, shear strength, or MOR. Polystyrene glulam had higher density and hardness, but lower moisture content and MOE. In other work, Hadi et al. [17] reported that polystyrene glulams of three fast-growing wood species had lower values for MOR and MOE, equal shear strength and wood failure, and higher hardness than the untreated glulam, and both glulams had slight delamination in the hot water test. In further research, Nurhanifah et al. [24] showed that the shear strength of sengon (Falcataria moluccana) solid wood was not significantly different from polystyrene glulam, meaning the impregnation process of styrene into sengon wood did not affect the gluing process. Furthermore, it was reported that the polystyrene-impregnated glulam had better resistance to termite attacks than solid wood, but its performance was not significantly different from that of the untreated glulam.

Reflecting on the previous research listed above, since polystyrene wood and its polystyrene glulam demonstrated better mechanical properties and resistance against termite attacks than untreated wood, the objective of this current study was to enhance the resistance of other tropical fast-growing tree species to subterranean termite attacks (Coptotermes curvignathus) through the manufacturing of glulam made from polystyrene wood laminae. The young tropical wood species studied were manii (Maesopsis eminii), mangium (Acacia mangium), and rubber-wood (Hevea brasiliensis).

## 2. Materials and Methods

### 2.1. Materials

Wood specimens for glulam manufacturing were sourced from plantation of people’s forests in the Bogor area, West Java, Indonesia. The log species were manii (Maesopsis eminii Engl.), mangium (Acacia mangium Willd.), and rubber-wood (Hevea brasiliensis Muell Arg.). All logs had a diameter of less than 20 cm and were cut from young stands of less than 10 years old. The logs were cut into flat-sawn timber for lamina manufacturing at 1.67 cm by 6 cm by 50 cm (in thickness, width, and length, respectively) and then kiln-dried to reach about a 12% moisture content. The recorded mechanical properties of the MOE and MOR of manii, mangium, and rubber-wood were 4.5 ± 0.4 GP and 42.7 ± 3.3 MP, 10.5 ± 1.5 GP and 79.4 ± 8.4 MP, and 6.1 ± 0.6 GP and 50.5 ± 8.6 MP, respectively. Meanwhile, the MOE and MOR of the glulam controls for manii, mangium, and rubber-wood were 6.8 ± 0.4 GP and 61.7 ± 4.9 MP, 12.1 ± 1.1 GP and 102.1 ± 9.0 MP, and 8.3 ± 0.4 GP and 72.7 ± 4.4 MP, respectively. Furthermore, the MOE and MOR of polystyrene glulam of manii, mangium, and rubber-wood were 5.7 ± 0.5 GP and 48.6 ± 3.2 MP, 11.3 ± 0.6 GP and 81.6 ± 7.6 MP, and 7.5 ± 0.9 GP and 64.9 ± 8.7 MP, respectively [11]. The styrene monomer and potassium peroxydisulfate used as catalysts were bought from TokoFRP and PT. Merck Indonesia Tbk, Jakarta, Indonesia.

### 2.2. Glulam Manufacturing and Its Properties

The laminae and glulam manufacturing process is shown in Figure 1. Prior to glulam manufacturing, the modulus of elasticity (MOE) for each lamina was estimated using a non-destructive testing system by means of a wood-grading device, a Panter MPK-5 made by IPB University, Bogor, Indonesia [25]. The laminae were then classified according to their MOE values. The laminae with higher MOE values were used for the outer layers in three-layered glulam manufacturing, while those with lower values were applied for the inner layers.

Polystyrene-impregnated laminae were prepared by weighing the laminae and exposing them to a vacuum at 600 mmHg for 30 min in a tank. For the impregnation process, potassium peroxydisulfate was added as a catalyst to styrene monomer (1:100 *v*/*v*), and the solution was introduced to the tank as the vacuum was released. Afterward, a pressure of 10 kg/cm^2^ was applied for another 30 min. After the impregnation process, each lamina was wrapped with aluminum foil and placed in an oven at 80 °C for 24 h. The foil was then removed, and each lamina specimen was weighed to calculate the polymer loading or weight percent gain (WPG). Conditioning of the specimens was conducted at room temperature for two weeks. In addition, three-layered glulam was manufactured using the laminae with a higher MOE in the face and back layers, while the lamina with the lowest MOE was used for the core layer. The laminae were placed with a longitudinal fiber orientation along the length of the glulam. Isocyanate glue was applied with a single spread glue line at 280 g/m^2^ [17], and the laminae were then cold-pressed with a specific pressure of 10 kg/cm^2^ for 3 h, followed by conditioning at room temperature for two weeks in the laboratory of the Centre for Standardization of Sustainable Forest Management Instruments, Ministry of Environment and Forestry, Bogor, Indonesia.

For comparison purposes, untreated glulam and solid wood specimens were also prepared. Six replications of the test specimens were manufactured for each treatment combination of wood species and wood products.

### 2.3. Physical Properties’ Determination

The physical properties measured were the density and moisture content, according to Japanese Agricultural Standard JAS 234-2003 [26].

### 2.4. Laboratory Test of Termite Attack

The Indonesian standard SNI 7207-2014 for subterranean termite attacks in laboratory tests [27] was used in this study. To begin, 200 g of sterilized sand with a moisture content of 7% under a water-holding capacity was placed in a glass container, then a wood test wood specimen was added to the container. The wood specimen was stood up almost vertically from the bottom of the container, touching its side. Two hundred healthy and active Coptotermes curvignathus Holmgren subterranean worker termites from a laboratory colony were added to each container. Schematically, the test unit is shown in Figure 2.

The containers were left in a dark room, at 25 °C to 30 °C and 80% to 90% relative humidity, for 4 weeks. The containers were weighed weekly, and if the moisture content of the sand decreased by 2% or more, water was added to achieve the moisture content standard.

At the end of the exposure period, the wood specimens were cleaned and then placed in an oven at 100 °C to reach their oven-dry weight. The endpoints evaluated were the wood density, moisture content (MC) of the wood, termite mortality, wood weight loss, wood resistance class based on the percentage of wood weight loss, and termite feeding rate, based on the work of Hadi et al. [18]. The protection level of wood against termite attacks was rated according to Table 1, following ASTM D 1758-06. 2006 [27].

Furthermore, based on the WL, the wood resistance class against subterranean termites could be classified by referring to SNI 7207-2014 [28], as shown in Table 2.

### 2.5. Prioritizing Wood Species and Wood Product

The wood species and wood product from each parameter or response were quantified, sorted, and scored numerically using the Likert scale [29], from low to high. A low score showed a lack of priority, and a higher score showed a better priority. Letters a, b, and c for the wood species, and p, q, and r for the wood product, were equal to the scores of one, two, and three, respectively. All letters referred to Duncan’s multi-range test result for each wood species and wood product. The total score was obtained by adding together all parameter scores, including the termite mortality, feeding rate, weight loss, resistance class, and protection level. A higher total score reflected a higher priority of wood species and wood product.

### 2.6. Data Analysis

The data were analyzed in a completely randomized block design using two factors, the wood species and wood product. The wood species, as a block factor, consisted of three levels, namely manii, mangium, and rubber-wood. The wood products factor consisted of three levels, too, namely the solid wood, control glulam, and polystyrene glulam. Duncan’s multi-range test was carried out for further analysis when the main factor was significantly different at *p* ≤ 0.05 [30].

## 3. Results and Discussion

### 3.1. Physical Properties

Physical properties—the weight percent gain (WPG), density, and moisture content (MC)—of each wood species and wood product are presented in Table 3. The summary of variance analysis is shown in Table 4, and Duncan’s multi-range test is described in Table 5. As can be seen in Table 4 and Table 5, the wood species highly significantly affected the WPG of polystyrene polymer loading. Rubber-wood had the smallest WPG due to having the highest density and was different from the other wood species, while the remaining two species were largely the same. These results were in line with Hadi et al. [22], who stated that a higher-density wood species produces a lower WPG because it has a smaller void. The WPG (10 to 21%) in this study was much lower compared to polystyrene-impregnated Polish wood (88 to 135%) [23].

Moreover, the wood density was highly significantly affected by not only the wood species but also the wood product (Table 4 and Table 5). Manii had the lowest density, followed by mangium and rubber-wood, with the three wood species significantly different from each other. The density values were in line with Martawijaya et al. [31]. In regard to wood products, solid wood had the lowest density, followed by control glulam and polystyrene glulam. The three products were significantly different from each other. Control glulam had a higher density than solid wood due to its glue-line presence and press treatment. The polystyrene glulam had the highest density because it had polystyrene impregnated in each lamina. Nevertheless, the WPG should be increased to achieve better physical properties.

The wood species and wood products did not affect the MC. All wood specimens had the same MC, typical of the Bogor area; as Kadir [32] stated, the MC varied from 12% to 18%. The MC of the wood products, meanwhile, varied from 10.4% to 11.8%, a value range that matched JAS 234-2003 [26].

### 3.2. Termite Test

The responses from the laboratory tests for termite resistance, including termite mortality, wood weight loss, wood resistance class, protection level, and termite feeding rate, are presented in Table 6. The analysis of variance outcome is shown in Table 4, Duncan’s multi-range test is summarized in Table 5, and images of the wood specimens after the test are shown in Figure 3.

According to the variance analysis in Table 4, termite mortality was affected by the wood species and wood products. A multi-range test, as presented in Table 5, showed that termite mortality on mangium (61.1%) was the highest, and different from manii (7.6%) and rubber-wood (8.5%), which were almost the same. Mihara et al. [33] also noted that mangium heart-wood contained flavonoids (2,3-trans-3,4′,7,8,-tetrahydroxyflavanone, teracacidin, 4′,7,8,-trihydroxyflavanon, and 3,4′,7,8,-tetrahydroxy-flavanone) that could resist fungal (*P. noxius* and *P. badius*) attacks. These findings may indicate that those flavonoids act as termiticides. Furthermore, this study found that the termite mortality on mangium wood was very high. When we consider classifications, Oey [34] noted that mangium belonged to termite resistance class III, while manii and rubber-wood belonged to class V [35,36]; in, the Indonesian standard [28] (see Table 2), class I is very resistant, while class V is very poorly resistant, to subterranean termite attack.

In terms of wood products, polystyrene glulam (38.5%) had the highest termite mortality, followed by control glulam (25.6%) and then solid wood (13.1%); the three wood products were significantly different from each other. Polystyrene glulam had the highest termite mortality because it had the highest density. This finding was in line with that of Arango et al. [37], who noted that higher wood density could have higher resistance to subterranean termite attack. Furthermore, the results were also in line with Hadi et al. [18], who noted that polystyrene wood had much higher termite mortality than solid wood.

Referring to the variance analysis in Table 4, the wood weight loss was significantly affected by the wood species and wood products. In the multi-range test results (see Table 5), mangium (10.0%) had the lowest weight loss, which was significantly different from manii (20.4%) and rubber-wood (19.3%), the two of which were almost the same. The results were in line with the termite mortality, with mangium wood again highest. Put simply, it had the fewest number of living termites to feed the wood, causing the wood weight loss to be the lowest.

In terms of the wood products, solid wood had the highest weight loss (23.8%), followed by control glulam (14.2%) and polystyrene glulam (11.7%), with the two glulams almost the same. These glulams had a higher density than solid wood, and they also had a glue line; both factors could have decreased the wood weight loss. The weight loss of polystyrene glulam was slightly lower than that of control glulam. However, statistically, they were not significantly different. This result matched that of Nurhanifah et al. [24], who reported that solid wood of sengon (*Falcataria moluccana*) was significantly less resistant than control glulam and polystyrene glulam, with the two glulams almost the same. To achieve the much lower weight loss of polystyrene glulam, we suggest that the weight gain of polystyrene on laminae should be increased. Referring to Hadi et al. [23], after a one-year exposure in the field, four Polish woods with polymer loading between 88 and 135% had a weight loss of 19%, while the untreated woods had 100% weight loss or totally failed. Nevertheless, the polystyrene weight gain of laminae should be considered to a certain degree, along with its optimal mechanical properties, as mentioned by Hadi et al. [17], who noted that the shear strength and rupture modulus of polystyrene glulam were lower than those of solid wood.

The weight loss reflected the resistance class of the wood in the laboratory test, as is described in Table 2. Referring to Table 4, the resistance class was affected by the wood species and wood products. Based on the multi-range test in Table 5, mangium wood had the highest resistance class, which was different from manii and rubber-wood, while those were almost the same. This finding was in line with that for the original or untreated wood species, confirming that mangium belongs to resistance class III, while the others belong to class V. In terms of the wood products factor, solid wood that belonged to the lowest class (average class 4.5) was different from control glulam and polystyrene glulam, with the two glulams were almost the same. This phenomenon was in line with the wood weight losses.

The protection level of the wood specimen dictated how much of the wood specimen was left compared to its original condition; the highest value of 10 indicated that the wood specimen was very resistant, while zero was for failure, as described in Table 1. The protection level of the wood specimen was affected by the wood species and wood products. Mangium wood, with a value of 8.3, was the most resistant, followed by rubber-wood (7.6) and manii (6.0); for this factor, the three wood species were significantly different from each other. These protection levels of the three wood species were in line with the weight loss found in this study. Regarding wood products, solid wood was the most susceptible to attack by termites, which was indicated by the lowest protection level (value of 5.8). The performance of solid wood was significantly different from control glulam (value of 7.5) or polystyrene glulam (value of 7.7), with the two glulams almost the same. These findings matched those of Hadi et al. [18], who noted that the control glulam had a lower wood weight loss than solid wood, and Hadi et al. [22], who reported that polystyrene wood had a lower wood weight loss and higher protection level than solid wood. In other words, the glulams were more resistant than the solid wood.

The wood consumption of each termite per day, or feeding rate, was highly affected by the wood species and wood products, as shown in Table 4. Referring to the multi-range test presented in Table 5, rubber-wood had the highest feeding rate (134.6 µg/termite/day), which was different from mangium (93.4 µg/termite/day) and manii (86.9 µg/termite/day), the two of which were almost the same. Rubber-wood belonged to the very poor resistance class or was very easily attacked by the termites (resistance class V, the lowest class in Indonesian standard), and it had the highest density (0.73 g/cm^3^); these findings indicate that the wood was very easily consumed by the termites, with high mass feeding.

If we look at the solid rubber-wood, the feeding rate in this study reached 147 ± 13 µg/termite/day, a value similar to that put forward by Arinana et al. [36], who reported that the termite feeding rate for rubber-wood was 129 ± 10 µg/termite/day. Likewise, the feeding rates of manii and mangium solid wood were 131 ± 4 µg/termite/day and 100 ± 10 µg/termite/day, respectively, which were similar to those found by Hadi et al. [18] (at 145 ± 41 µg/termite/day and 122 ± 104 µg/termite/day, respectively). In other words, the feeding rates for solid wood in this work were similar to the findings of other works.

According to Table 5, the feeding rate for solid wood was the highest (125.8 µg/termite/day) and significantly different from those of control glulam (100.0 µg/termite/day) and polystyrene glulam (89.0 µg/termite/day), which also differed from one another. This finding was in line with the wood weight loss, where solid wood had the highest weight loss, followed by control glulam and polystyrene glulam.

### 3.3. Priority Product

According to Table 7, in terms of the wood species, mangium wood should be prioritized. In terms of wood products, meanwhile, polystyrene glulam should be prioritized, followed by control glulam and then wood solid. These priorities were decided based on the classification of the results for specific critical parameters in Table 5. A parameter that positively affects the treatment (e.g., termite mortality, protection level) improves the priority value, while a parameter that negatively affects the treatment (e.g., weight loss, resistance class, and feeding rate) decreases the priority value. Thus, the highest weight loss values due to termite attacks lower the priority values, while the highest termite mortality values raise them.

## 4. Conclusions

From the discussion above, it can be concluded that mangium wood has a better resistance class (class III, moderately resistant) to subterranean termite attacks than manii and rubber-wood, which belong to the lowest class (class V, very poorly resistant) of Indonesian standard SNI 7207-2014. In terms of wood products, solid wood has the lowest resistance class (class V) to subterranean termite attacks when compared with control glulam and polystyrene glulam (both class IV, poor resistance); the glulams were equal and had a better resistance than solid wood. To enhance polystyrene glulam’s resistance to termite attacks, the polymer loading of polystyrene on each lamina should be increased to achieve optimal mechanical properties, including the modulus of elasticity, modulus of rupture, and shear strength. Of the wood products studied, polystyrene glulam should be the most prioritized since it demonstrated the best resistance to subterranean termite attacks.

## Figures and Tables

**Figure 1 polymers-14-04003-f001:**
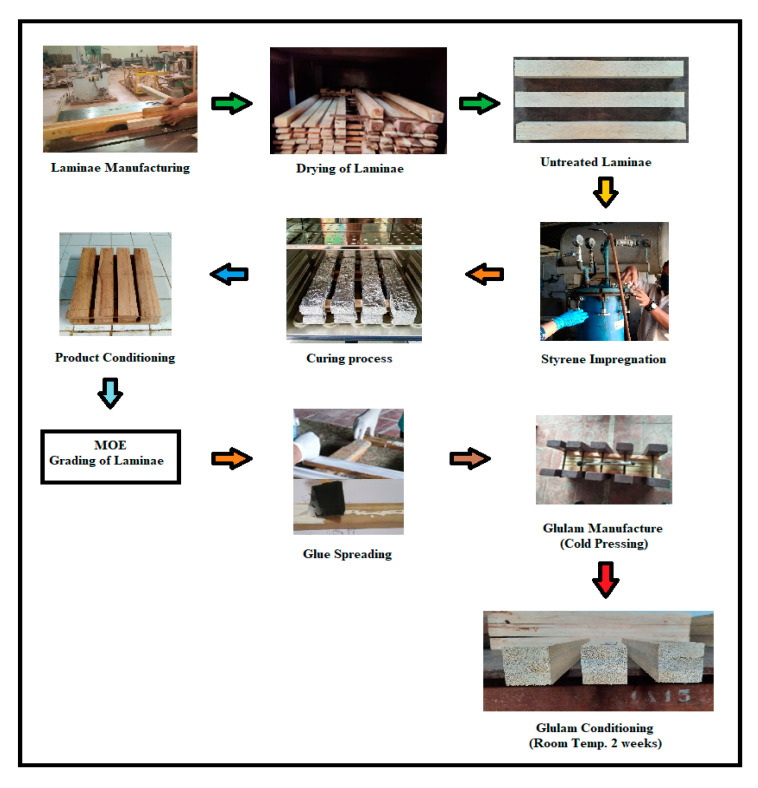
Laminae and glulam manufacturing scheme.

**Figure 2 polymers-14-04003-f002:**
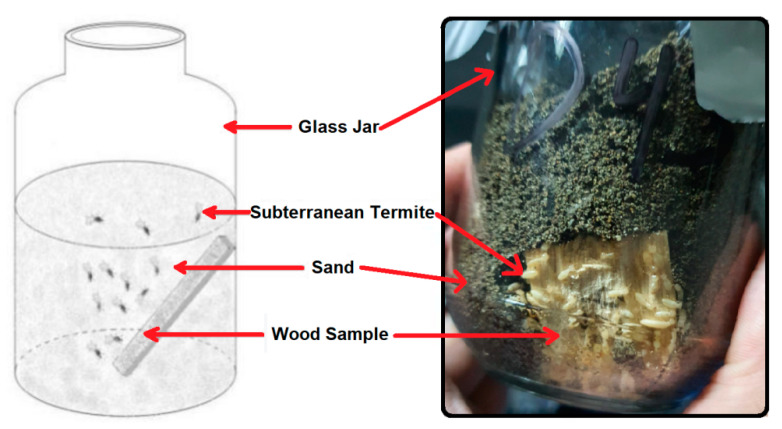
Test unit of subterranean termite attack on the wood specimen.

**Figure 3 polymers-14-04003-f003:**
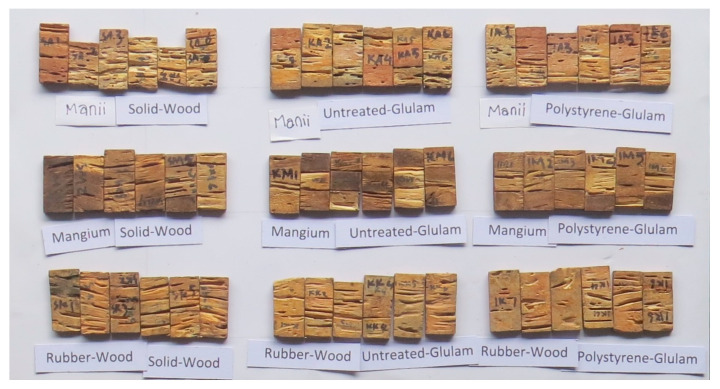
Wood specimens after the laboratory test.

**Table 1 polymers-14-04003-t001:** Rating system of protection level against termite attack.

Rating	Criteria
10	no attack; 1 to 2 small nibbles permitted
9	nibbles to 3% of cross-section
8	penetration of from 3 to 10% of cross-section
7	penetration of from 10 to 30% of cross-section
6	penetration of from 30 to 50% of cross-section
4	penetration of from 50 to 75% of cross-section
0	failure

**Table 2 polymers-14-04003-t002:** Resistance class against subterranean termite attack.

Resistance Class	Sample Condition	Mass Loss (%)
I	Very resistant	<3.52
II	Resistant	3.52–7.50
III	Moderately resistant	7.50–10.96
IV	Poorly resistant	10.96–18.94
V	Very poorly resistant	>18.94

**Table 3 polymers-14-04003-t003:** Physical properties of wood species and wood products.

Wood Species	Wood Product	WPG (%)	Density (g/cm^3^)	MC (%)
Manii	Solid	0	0.42 (0.03)	11.64 (0.51)
	Control glulam	0	0.43 (0.01)	10.43 (1.04)
	Polystyrene glulam	18.2 (4.2)	0.49 (0.01)	10.67 (1.54)
Mangium	Solid	0	0.62 (0.03)	11.74 (3.3)
	Control glulam	0	0.66 (0.03)	11.72 (0.35)
	Polystyrene glulam	21.1 (1.6)	0.75 (0.01)	11.03 (0.32)
Rubber-wood	Solid	0	0.70 (0.05)	11.78 (0.32)
	Control glulam	0	0.73 (0.01)	11.59 (0.33)
	Polystyrene glulam	10.4 (3.7)	0.77 (0.02)	11.57 (0.26)

Note: Values in parentheses are standard deviations.

**Table 4 polymers-14-04003-t004:** Variance analysis of physical properties and laboratory termite test.

Parameter	Wood Species	Wood Product
WPG	**	-
Density	**	**
Moisture content	NS	NS
Weight loss	**	**
Resistance class	**	**
Protection level	**	**
Mortality	**	**
Feeding rate	**	**

Notes: ** highly significant (*p* ≤ 0.01); NS not significant (*p* > 5%).

**Table 5 polymers-14-04003-t005:** Duncan’s multi-range test of physical properties and laboratory termite test.

Parameter		WPG	Density	MC	Termite Mortality	Weight Loss	Resistant Class	Protection Level	Feeding Rate
Wood species	Manii	18.2 b	0.45 a	11.1 a	7.6 a	20.4 a	4.3 a	6.0 a	86.9 b
	Mangium	21.1 b	0.68 b	11.5 a	61.1 b	10.0 b	3.0 b	8.3 c	93.4 b
	Ruber-wood	10.4 a	0.73 c	11.8 a	8.5 a	19.3 a	4.5 a	6.7 b	134.6 a
Wood products	Solid	-	0.58 p	11.2 p	13.1 p	23.8 p	4.5 p	5.8 p	125.8 p
	Control glulam	-	0.61 q	11.4 p	25.6 q	14.2 q	3.8 q	7.5 q	100.0 q
	Polystyrene glulam	-	0.67 r	11.7 p	38.5 r	11.7 q	3.5 q	7.7 q	89.0 r

Note: The same letters in a column are not significantly different (*p* ≤ 0.05).

**Table 6 polymers-14-04003-t006:** Weight loss, resistance class, attack degree, mortality, and feeding rate of laboratory test.

Wood Species	Wood Products	Mortality (%)	Weight Loss (%)	Resistance Class	Protection Level	Feeding Rate (µg/Termite/Day)
Manii	Solid	7.2 (0.8)	34.6 (5.0)	4.8 (0.4)	4.0 (0.0)	130.8 (4.1)
	Control glulam	7.3 (1.8)	13.3 (1.6)	4.0 (0.0)	7.0 (0.0)	64.9 (6.4)
	Polystyrene glulam	7.8 (1.7)	13.2 (1.2)	4.0 (0.0)	7.0 (0.0)	65.0 (3.6)
Mangium	Solid	24.7 (3.9)	15.2 (1.5)	3.7 (1.5)	7.0 (0.0)	99.8 (10.4)
	Control glulam	61.7 (11.2)	8.2 (1.2)	3.0 (0.6)	9.0 (0.0)	90.4 (9.7)
	Polystyrene glulam	96.9 (3.8)	6.6 (1.1)	2.3 (0.5)	9.0 (0.0)	90.0 (4.2)
Rubber-wood	Solid	7.4 (1.9)	21.7 (2.2)	5.0 (0.0)	6.5 (1.2)	146.8 (13.0)
	Control glulam	7.9 (1.8)	20.1 (3.4)	4.3 (0.5)	6.5 (1.2)	144.8 (4.5)
	Polystyrene glulam	7.8 (1.6)	15.2 (2.9)	4.2 (0.4)	7.0 (0.0)	112.1 (15.2)

Note: Values in parentheses are standard deviations.

**Table 7 polymers-14-04003-t007:** Prioritizing wood species and wood products.

Parameter		Termite Mortality	Weight Loss	Resistance Class	Protection Level	Feeding Rate	Sum
Wood sp.	Manii	1	1	1	1	2	6
	Mangium	2	2	2	3	2	11
	Rubber-wood	1	1	1	2	1	6
Wood products	Solid	1	1	1	1	1	5
	Control glulam	2	2	2	2	2	10
	Polystyrene glulam	3	2	2	2	3	12

Note: Numbering came from scoring based on Table 5, with a and p as 1, b and q as 2, and c and r as 3.

## Data Availability

Not applicable.

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
