# Peer review of "Polystyrene-Impregnated Glulam Resistance to Subterranean Termite Attacks in a Laboratory Test"

_polymers, 2022, doi:10.3390/polym14194003_

Round 1
Reviewer 1 Report
In this paper, the authors have studied the impact of adding/coating various wood samples with polystyrene on termite attacks. This study is useful for the public, as termite is a huge problem in Asia. Overall, this study is meaningful, but I have the comments below that should be addressed before this paper is accepted:
1. Please mention the equivalent ASTM or ISO standards where Indonesian and Japanese standards had been mentioned.
2. Please include the pictures and schematic of the sample-making process.
3. What is the reason for the higher density of control glulam over solid samples (Table 3).
Reviewer 2 Report
The work presented for review addresses the ever-present problem of an insufficient raw material base in the wood industry. For many years, the development of the global economy has involved an increasing demand for products made from wood. Although wood is a renewable material and even fast-growing species are found in large quantities in some regions of the world, the industry is constantly looking for new resources. This is because some of the available raw material is of poor quality. Therefore, various efforts are being made to improve the quality of this raw material. The authors' research aimed to increase the resistance of fast-growing tropical trees to termites. The subject matter undertaken is up-to-date and in line with global scientific trends. The authors have made a correct introduction to the research topic. However, the objective presented at the end of the section is little understood. The authors gave the purpose of the study much better in the Abstract than in the introduction.
The methodology of the study is correctly described. Although it does not affect the paper's main purpose, the authors could have given the range of MOE of the wood used in the study. It also seems that the formulae 1-3 and 5 given are quite obvious and could be omitted. I leave the decision to the authors.
The authors clearly present the results of their study. During the discussion, they refer to the results of other authors.
Author Response
Reviewer-2
The work presented for review addresses the ever-present problem of an insufficient raw material base in the wood industry. For many years, the development of the global economy has involved an increasing demand for products made from wood. Although wood is a renewable material and even fast-growing species are found in large quantities in some regions of the world, the industry is constantly looking for new resources. This is because some of the available raw material is of poor quality. Therefore, various efforts are being made to improve the quality of this raw material.
The authors' research aimed to increase the resistance of fast-growing tropical trees to termites. The subject matter undertaken is up-to-date and in line with global scientific trends. The authors have made a correct introduction to the research topic. However, the objective presented at the end of the section is little understood. The authors gave the purpose of the study much better in the Abstract than in the introduction.
Author’s Comment:
The introduction, particularly the objective of this research, has been revised. In the text it is written as follows:
Reflecting on the previous research above, since the polystyrene wood and its polystyrene glulam provided better mechanical properties and resistance against termite attacks than the untreated wood and glulam control, therefore, the objective of this current study was to enhance the resistance of other tropical fast-growing tree species against subterranean termite (Coptotermes curvignathus) attacks through the manufacturing of glulam made from polystyrene wood laminae. These young tropical wood species studied were manii (Maesopsis eminii), mangium (Acacia mangium), and rubber-wood (Hevea brasiliensis).
The methodology of the study is correctly described. Although it does not affect the paper's main purpose, the authors could have given the range of MOE of the wood used in the study.
Author’s Comment:
The MOE/MOR values of each wood species and its glulam control and plystyrene glulam have been added into the text. In the text, this mechanical properties of the material were written as follows:
The recorded mechanical properties as MOE and MOR of manii, mangium, and rubber-wood were 4.5±0.4 GPa and 42.7±3.3 MPa; 10.5±1.5 GPa and 79.4±8.4 MPa; 6.1±0.6 GPa and 50.5±8.6 MPa, respectively. Whereas, the MOE and MOR of glulam control of manii, mangium, and rubber-wood were 6.8±0.4 GPa and 61.7±4.9 MPa; 12.1±1.1 GPa and 102.1±9.0 MPa; 8.3±0.4 GPa and 72.7±4.4 MPa, respectively. Further, the MOE and MOR of polystyrene glulam of manii, mangium, and rubber-wood were 5.7±0.5 GPa and 48.6±3.2 MPa; 11.3±0.6 GPa and 81.6±7.6 MPa; 7.5±0.9 GPa and 64.9±8.7 MPa, respectively (Hadi et al. 2015).
It also seems that the formulae 1-3 and 5 given are quite obvious and could be omitted.
Author’s Comment:
The formulae 1 - 5 have been omitted. In the text, some of formulae referred to Hadi, Y.S.; Massijaya, M.Y.; Hermawan, D.; Arinana, A. Feeding rate of termites in wood treated with borax, acetylation, polystyrene, and smoke. J. Indian Acad. Wood Sci. 2015, 12, 74-80., while another one referred to Thybring, E.E. The decay resistance of modified wood influenced by moisture exclusion and swelling reduction. Int. Biodeterior Biodegrad. 2013, 82, 87–95.
I leave the decision to the authors.
The authors clearly present the results of their study.
During the discussion, they refer to the results of other authors.
Reviewer 3 Report
Hadi and coworkers reported a study on the Polystyrene-impregnated glulam resistance to subterranean termite attacks in a laboratory test. Though, not much improvement in the resistance to subterranean termite was seen, however, the work is novel, well planned and compared with the literature. Therefore, recommended for acceptance in the current form.
Author Response
Reviewer-3
Hadi and coworkers reported a study on the Polystyrene impregnated glulam resistance to subterranean termite attacks in a laboratory test. Though, not much improvement in the resistance to subterranean termite was seen, however, the work is novel, well planned and compared with the literature. Therefore, recommended for acceptance in the current form.
Author’s Comment:
Thank you so much for your recommendation. The work will be continued to get a higher weight percent gain (WPG) of polystyrene into the wood and to get better results, we do hope so.
